# Functional Analysis of V2 Protein of Beet Curly Top Iran Virus

**DOI:** 10.3390/plants11233351

**Published:** 2022-12-02

**Authors:** Atiyeh Bahari, Araceli G. Castillo, Naser Safaie, Eduardo R. Bejarano, Ana P. Luna, Masoud Shams-Bakhsh

**Affiliations:** 1Plant Pathology Department, Faculty of Agriculture, Tarbiat Modares University (TMU), Tehran 336-14115, Iran; 2Consejo Superior de Investigaciones Científicas (IHSM-UMA-CSIC), Departamento Biología Celular, Genética y Fisiología, Universidad de Málaga, 29010 Malaga, Spain

**Keywords:** beet curly top disease, PTGS suppressor, hypersensitive response-like, pathogenicity, systemic movement

## Abstract

Geminivirus beet curly top Iran virus (BCTIV) is one of the main causal agents of the beet curly top disease in Iran and the newly established *Becurtovirus* genus type species. Although the biological features of known becurtoviruses are similar to those of curtoviruses, they only share a limited sequence identity, and no information is available on the function of their viral genes. In this work, we demonstrate that BCTIV V2, as the curtoviral V2, is also a local silencing suppressor in *Nicotiana benthamiana* and can delay the systemic silencing spreading, although it cannot block the cell-to-cell movement of the silencing signal to adjacent cells. BCTIV V2 shows the same subcellular localization as curtoviral V2, being detected in the nucleus and perinuclear region, and its ectopic expression from a PVX-derived vector also causes the induction of necrotic lesions in *N. benthamiana*, such as the ones produced during the HR, both at the local and systemic levels. The results from the infection of *N. benthamiana* with a V2 BCTIV mutant showed that V2 is required for systemic infection, but not for viral replication, in a local infection. Considering all these results, we can conclude that BCTIV V2 is a functional homologue of curtoviral V2 and plays a crucial role in viral pathogenicity and systemic movement.

## 1. Introduction

RNA silencing is one of the most efficient antiviral defences in plants. It is activated by a viral double-stranded RNA (dsRNA) that is then processed by Dicer-like (DCL) ribonucleases producing 21 to 24-nt primary virus-derived small interfering RNAs (vsiRNAs). Subsequently, in the amplification step, host-encoded RNA-dependent RNA polymerases (RDRs) synthesize dsRNA from viral single-stranded RNA producing secondary vsiRNAs that, together with the primary vsiRNAs, are responsible for the spreading of the silencing through the plant. vsiRNAs bind to different Argonaute (AGO)-containing effector complexes and provide targeting specificity for RNA dicing (PTGS or post-transcriptional gene silencing) or chromatin modification (TGS or transcriptional gene silencing). Viruses counterattack by producing one or several proteins that suppress this defence by targeting different steps in the pathway [1,2,3,4].

Geminiviruses are a group of insect-transmitted plant viruses that cause destructive diseases in major crops worldwide. The virions of this family are twinned and contain a single copy of circular single-stranded DNA, ranging in size from 2.5 to 3.0 kb [5,6,7].

During infection, geminiviruses must confront both types of silencing, PTGS, and TGS. As counter-defence, they encode more than one viral RNA silencing suppressor, being that the V2 proteins exhibit the most efficient PTGS suppression activity [8,9].

V2 (AV2) is a multifunctional protein that has been described as: a (i) PTGS or a TGS suppressor in most of the geminivirus species tested to date [10,11,12,13,14,15,16,17,18,19]; (ii) it is also required for viral movement and spreading of the virus through the plant [4,20,21,22]; and (iii) it is involved in the regulation of other host defence responses different than RNA silencing [23,24,25].

Beet curly top Iran virus (BCTIV) is one of the causal agents of beet curly top disease (BCTD) in Iran, described recently as the type species of the genus *Becurtovirus* [26,27]. Its genome presents a unique organization: similarly to *Mastrevirus*, it contains two intergenic regions, one large (LIR) and another small (SIR), located at opposite sides of the genome. BCTIV LIR includes a sequence capable of forming a stem-loop structure and a novel nonanucleotide (TAAGATT/CC) with a unique nick site. The BCTIV genome comprises three virion-sense (V1, V2, and V3) and two complementary-sense (C1 and C2) ORFs. Based on a comparison of nucleotide sequence identity of individual genes, the three virion-sense ORFs V3, V2 and V1 are similar to their positional homologs in the curtoviruses, whereas C1 and C2 are more similar to the ones from mastreviruses [28,29] (Figure 1A). 

This work aims to determine the role of BCTIV V2 in pathogenicity and its function as an RNA silencing suppressor. This knowledge will help to study further the molecular mechanisms involved in the infection process and to develop strategies for decreasing the impact of this disease in agriculture.

## 2. Results

### 2.1. BCTIV V2 Aminoacidic Sequence Analysis

Amino acidic comparison among V2 from BCTIV with the homolog proteins from curtoviruses and begomoviruses revealed that BCTIV V2 contains a conserved CK2/PKC (protein kinase CK2/protein kinase C) phosphorylation motif (hereafter named P1), present in all V2 proteins and two putative CK2 and PKC phosphorylation motifs. These were predicted in BCTV V2 (named P2 and P3, respectively) and are also present in BCTIV V2, but not in begomoviral proteins (Figure 1B). Besides those phosphorylation motives, all V2 proteins have similar hydrophobic profiles displaying two hydrophobic regions at the N-terminus (hereafter named H1 and H2), followed by a long hydrophilic region at the C-terminus (Figure 1B). These hydrophobic regions, along with P1, but not P2 or P3, have been proved to be essential for PTGS suppression activity and pathogenicity in the *Begomovirus* and *Curtovirus* genera [25,30]. Besides those conserved regions, the sequence alignment analysis identified a nuclear localization signal (NLS) present in BCTIV V2, as well as in V2 from the curtovirus BCTV protein sequences, while it was absent in most of the begomoviruses. (Appendix A) [31,32].

### 2.2. BCTIV V2 Is a Local PTGS Suppressor

To test if BCTIV V2 functions as a PTGS suppressor, we carried out transient expression assays in wt *N. benthamiana* plants. Leaves were co-infiltrated with *Agrobacterium* cultures expressing GFP (GFP) and BCTIV V2 (thereafter named IVV2). As controls, leaves were co-infiltrated with GFP, and the empty vector (EV) (negative control) or a plasmid expressing the viral silencing suppressor V2 from BCTV (thereafter named BCV2) (positive control) were considered. Leaves were observed under UV light at 6 dpi. Leaves co-infiltrated with GFP and IVV2 displayed high intensity of GFP fluorescence, as leaves co-infiltrated with GFP and BCV2. Conversely, in the patches infiltrated with the empty vector, the intensity of fluorescence had disappeared almost completely (Figure 2A). These differences in green fluorescence were confirmed by Western blot analysis using an anti GFP antibody. GFP protein accumulation in tissues, co-infiltrated with GFP and IVV2 or BCV2 at 6 dpi, was considerably higher than in tissues co-infiltrated with GFP and the empty vector (l (Figure 2B)). Similarly, GFP mRNA also accumulates at a higher level in patches co-infiltrated with GFP and IVV2 or BCV2 compared to the ones co-infiltrated with GFP and the empty vector (Figure 2C). On the contrary, there were no significant differences in GFP siRNA accumulation between the different experimental conditions (Figure 2D). Expression of viral genes in the infiltrated tissues was confirmed by semiquantitative RT-PCR (Figure 2E).

### 2.3. BCTIV V2 Causes a Delay in the Long-Distance Spread of RNA Silencing

To determine the effect of BCTIV V2 on systemic PTGS, the constructs expressing GFP and IVV2 or BCV2 were agroinfiltrated in leaves of transgenic 16c *N. benthamiana* plants. Co-infiltration with GFP and P19 [33] or the empty vector were used as negative or positive controls, respectively.

Fluorescence was monitored under UV light to detect the initiation of systemic silencing in newly emerging leaves, from 4 to 30 dpi. To evaluate the silencing progress, we applied the arbitrary silencing index, described in ref [34] (Figure 3B). Systemic silencing in emerging upper leaves was not observed until 8 dpi. After 18 dpi, systemic GFP silencing was observed in plants co-infiltrated with GFP and the empty vector and in plants infiltrated either with IVV2 or BCV2, suggesting neither of these proteins are able to suppress completely the spreading of the GFP silencing. As expected, in plants infiltrated with P19, no systemic silencing was observed. However, there was a clear reduction of the number and the area of leaves that showed GFP silencing in the IVV2- and BCV2-infiltrated plants compared to ones infiltrated with the empty vector. This suggests that both proteins, IVV2 and BCV2, produce a delay in the spreading of the GFP silencing, although none of them can block it completely as P19 does. This delay was maintained until 30 dpi (Figure 3A,B).

### 2.4. BCTIV V2 Cannot Suppress Short-Range (Cell-to-Cell) Spread of Gene Silencing

Agroinfiltrated 16c plants were also employed to study the effect of BCTIV V2 on the short-range movement of the RNA silencing. For that, GFP expression was monitored with UV light in the cells surrounding the infiltrated tissues at 6 dpi to check if the silencing signal triggered by the GFP overexpression had been able to exit from the infiltrated area and cause the appearance of a red ring around the patch [35]. In the plants co-infiltrated with GFP and the empty vector, the red ring was observed at 6 dpi (Figure 3B). This red ring was also present around the patches infiltrated with BCV2, but not in those infiltrated with P19, as previously described [16,35]. In the plants infiltrated with IVV2, a red ring was also visible at 6 dpi, showing that this viral protein cannot block the cell-to-cell movement of the silencing signal to adjacent cells.

### 2.5. BCTIV V2 Triggers HR-like Response When Expressed from a PVX Derived Vector

As mentioned above, ectopic expression of V2 proteins from begomoviruses and curtoviruses from a potato virus X (PVX)- derived vector causes localized cell death in the infiltrated tissues and produces systemic necrosis associated with a hypersensitive response-like (HR-like) phenotype in *N. benthamiana* [11,15,16,24,34]. To study the possible role of BCTIV V2 in pathogenicity, we used the PVX-derived vector pGR107, to express the viral protein in wt *N. benthamiana* plants (PVX-IVV2). Agroinfiltration with the empty PVX vector was used as a negative control, whereas infiltration with PVX expressing either BCTV-V2 (PVX-BCV2) or the nonstructural protein (NSs) from tomato spotted wilt virus (PVX-NSs) were used as positive controls [16,36,37].

At 5 dpi, tissues infiltrated with the empty vector developed a local yellowing, typical for PVX, while tissues infiltrated with PVX-IVV2 showed intense local necrosis, similar to the infiltrated with the positive controls: PVX-NSs and PVX-BCV2 (Figure 4A). At 16 dpi, plants inoculated with PVX expressing any of the viral proteins, including IVV2, showed severe systemic necrosis and posteriorly did not recover from a viral infection, while the ones inoculated with the empty PVX vector were almost asymptomatic. Interestingly, although the three viral proteins induce necrosis, the intensity of the symptoms is more severe for PVX-NSs and PVX-IVV2 (Figure 4B).

### 2.6. BCTIV V2 Localizes in the Nucleus and Perinuclear Region

To obtain additional information on the function of BCTIV V2, we examined the subcellular localization of the protein, expressing GFP-fused versions of BCTIV V2 (both amino- and carboxyl-terminal fusions: GFP-IVV2 and IVV2-GFP, respectively) in wt *N. benthamiana* leaves. BCTV V2 fused to GFP in the amino-terminal region of the viral protein (GFP-BCV2), and free GFP (GFP) were used as controls. Thirty-six hours after the agroinfiltration, tissues were collected and visualized using a confocal microscope. BCTIV V2 seems to have a similar subcellular localization to other V2 proteins, including its curtoviral counterpart [16,38,39,40,41,42]. Accumulation of the protein is observed in the nucleoplasm and in the cell periphery, where it forms punctate fluorescent bodies. A closer examination shows that V2 accumulates in a discrete nuclear structure (Figure 5, Appendix A).

### 2.7. BCTIV V2 Is Essential for Systemic, but Not for Local Infection of BCTIV

To complete the characterization of BCTIV V2, we determined its relevance for the systemic viral infection. To achieve this, we generated a BCTIV infectious clone containing a stop mutation in the eighth codon of V2 ORF. This mutation also produced a change in the overlapping V3 gene, replacing a non-conserved arginine in the 33th position to leucine (Appendix A). Wt *N. benthamiana* plants were agroinoculated with the V2 mutant BCTIV (BCTIV-V2stp) or with the wild-type BCTIV (wt BCTV) as a positive control. Agroinoculation, with the empty vector, was used as a negative control. Symptoms were detectable in plants infected with wt BCTIV at 10 dpi, while plants infected with the V2 mutant remained did not show any symptom. At 28 dpi, wt BCTIV-infected plants showed the typical BCTIV symptoms (stunting, leaf curling, vein swelling, etc.), whereas those agroinoculated with BCTIV-V2stp did not develop any symptoms (Figure 6A).

To determine if the absence of symptoms correlated with an absence of viral DNA, total DNA was extracted from the newly emerging leaves of the inoculated plants at 28 dpi, and viral DNA was quantified by quantitative real-time PCR (qPCR). No BCTIV DNA was detected in the apical leaves of plants inoculated with V2 mutant BCTIV (Figure 6B), indicating that V2 mutant BCTIV could not infect plants systemically. To test if V2 is required for viral replication, we performed a local infection assay. Total DNA was extracted at 6 dpi from the infiltrated leaves, and viral DNA was quantified by qPCR. No differences in the amount of viral DNA accumulated in the infected tissues were detected in the leaves infected by wt BCTIV or V2 mutant BCTIV (BCTV-V2stp) (Figure 6C). This result showed that BCTIV V2 is not required for viral replication and points to a role in viral movement, as it has been described for other V2 proteins. However, considering that the mutation also affects V3, we cannot exclude the possibility that the lack of movement in the systemic infection is not only due to the mutation in V2.

## 3. Materials and Methods

### 3.1. Microorganisms, Plant Material, and Growth Conditions

Bacterial transformation and plasmid purification were carried out according to standard methods [43]. *Escherichia coli* strain DH5-α was used for subcloning. The *Agrobacterium tumefaciens* GV3101 strain was used for the agroinfiltration and agroinoculation/infection assays.

Plants used in this study were wild-type *Nicotiana benthamiana* and 16c *N. benthamiana* line (transgenic plants constitutively expressing green fluorescent protein [GFP] [44]. Plants were grown in chambers at 24 °C in long-day conditions (16 h light/8 h dark) before and after agroinfiltration/infection.

### 3.2. Sequence Analysis

The ClustalW algorithm was used to align V2 and V3 homolog proteins. The prediction of the nuclear localization signals was made by the NLS Mapper online tool (https://nls-mapper.iab.keio.ac.jp/cgi-bin/NLS_Mapper_form.cgi, accessed on 12 April 2022).

### 3.3. Plasmids and Cloning

The primer sequences used in this study are listed in Appendix A. All PCR-amplified fragments cloned in this work were fully sequenced.

Generation of protein expression clone: Gateway-compatible oligonucleotide primers BIV2Fw and BIV2RvSt or BIV2RvNoSt were used to amplify BCTIV V2 full-length ORF, with or without stop codon, respectively, and to generate entry clones by performing BP recombination reactions between the pDONR™ vector (Gateway™ pDONR™/Zeo Vector) and the attB PCR products. Subsequently, these entry clones were used for LR recombination reactions with Gateway™ destination binary vectors: pGWB2 yielding pGWB2-IVV2, for their expression in plants from a 35S promoter, and pGWB5 and pGWB6 yielding pGWB5-IVV2 and pGWB6-IVV2, to express V2 fused to GFP in the carboxyl- or the amino terminal region of the viral protein respectively [45].

Generation of PVX expression vectors: To generate the PVX expression vector, BCTIV V2 complete ORF was amplified using primers *Cla*I V2 F and *Sal*I V2 R, and they were subcloned into the pGEMT-easy vector (Promega Corp., Madison, WI, USA) to yield pGEMT- IVV2. Restriction fragments *Cla*I-*Sal*I from this plasmid containing the complete ORF were cloned into PGR107 [46], yielding the corresponding PVX expression vector PVX-IVV2.

Generation of V2 mutant BCTIV virus: As a first step, a 1.3 mer genome-length infectious BCTIV clone was obtained from the previously constructed BCTIV dimer infectious clone (accession number: JQ707949) [8]. The dimer was first digested by *Kpn*I (TaKaRa, Kyoto, Japan) to release a monomer, and subsequently with *EcoR*I and *EcoR*V (TaKaRa, Kyoto, Japan) to release a 0.3 mer fragment. This 0.3 mer fragment was cloned into *Kpn*I/*EcoR*I sites of pGreen0229 binary vector [46], yielding pGreen 0.3 BCTIV. Finally, the monomer obtained by *Kpn*I digestion was cloned into the *Kpn*I site of pGreen 0.3 BCTIV, to obtain the 1.3 mer genome-length infectious clone pGreen 1.3 BCTIV.

BCTIVV2 mutant virus was then generated by site-directed mutagenesis using the Nyztech Site-Directed Mutagenesis kit (Nyztech, Lisboa, Portugal) [47] with specific primers IVV2 stop F and IVV2 stop R (Appendix A), designed to introduce a stop codon in the V2 gene.

### 3.4. Agroinfiltration and Infection Assays

For local and systemic PTGS suppression assays and PVX infection, wild-type (wt) and 16c *N. benthamiana* leaves were agroinfiltrated, as previously described [34]. A long-wave UV lamp was used to detect GFP fluorescence (Black Ray model B 100 AP, Upland, CA, USA). Pictures were taken using an EOS 5D Canon digital camera (Tokyo, Japan)

For BCTIV infection, wt *N. benthamiana* plants at two leaf stage were agroinoculated, as described by [48], Elmer et al., 1988 (final OD600 = 0.2).

### 3.5. Subcellular Localization

For subcellular localization, *A. tumefaciens* was transformed with binary vectors containing BCTIV V2 or BCTV V2 fused to GFP in the carboxyl- or the amino terminal region of the viral protein. Wt *N. benthamiana* leaves were agroinfiltrated with cultures at OD600 0.5–1. Fluorescence was detected in epidermal cells 36 h after infiltration using a confocal microscope (Zeiss LSM 880).

### 3.6. Analysis of Nucleic Acids and Proteins

For BCTIV replication and infection analyses, plant DNA was extracted from the infiltrated (local) or the apical (systemic) leaves of the infected plants at 6 or 28 days post-infiltration (dpi), respectively. Plant DNA extraction was carried out following the method using CTAB buffer: 100 mg of agroinfiltrated tissue was homogenized into 500 μL of extraction buffer (2% cetyl trimethylammonium bromide (CTAB), 1.5 M NaCl, 100 mM Tris pH 8, 100 mM EDTA, pH 8) and incubated at 65 °C for 15 min. After cooling, the mixture was extracted with chloroform/isoamilic acid (24:1), and the nucleic acids were precipitated with isopropanol. DNA was finally resuspended in water and treated with RNase (10 mg/mL) (invitrogen). This DNA was digested with *Dpn*I to remove bacterial DNA in the infiltrated tissues (local infection) and then was subjected to qPCR analysis using primers BCTIV RTF and BCTIV RTR primers (Appendix A), and *N. benthamiana Elongation Factor α* (*EF1 α*) as normalizer [49]. A bacterial DNA preparation from an *Agrobacterium* containing the BCTIV infectious clone was used as a negative control.

Expression of BCTIV V2 and BCTV V2 in agroinfiltrated tissues was determined by semiquantitative RT-PCR. Total RNA was isolated using Trizol reagent (invitrogen) from the infiltrated leaves, treated with RNase-free Turbo DNaseI (Ambion), cleaned up by a phenol:chloroform treatment, and subjected to reverse transcription using Super-Script II RT reagent (Invitrogen) and oligo(dT) primers, according to the manufacturer’s instructions. Synthesized cDNA was subjected, then, to PCR amplification using viral specific primers (Appendix A) (initial denaturation was 95 °C for 5 min; 30 cycles: 95 °C for 45 s, 58 °C for BCTIV V2/53 °C for BCTV V2 and *EF1 α* for 30 s, 72 °C for 45 s; final extension 72 °C for 5 min). *EF1 α* was used as an internal gene control.

For high-molecular weight and small RNA Northern blots, total RNA was extracted using Trizol reagent (invitrogen) from the infiltrated leaves, and it was used for hybridizations as described ([50]). Probe labelling for U6 and GFP siRNA detection was performed, as described in ([51]).

For Western blot analysis, 120 mg of leaf tissue per sample was used. Total protein was extracted by 2X Laemmli buffer and separated using sodium dodecyl sulfate (SDS)–10% polyacrylamide gel electrophoresis (PAGE), then transfered to a PVDF membrane (Immobilon-P, Millipore, MA, USA) by a semi-dry electrotransfer (Bio-Rad, Hercules, CA, USA) [36]. It was then probed by an anti-GFP mouse monoclonal antibody (1:600, clone B-2; sc-9996, Santa Cruz Biotechnology, Dallas, TX, USA). Anti-mouse IgG conjugated by peroxidase (A9044, Sigma-Aldrich, Missouri, United States) was used as a secondary antibody at 1:80,000 (*V*/*V*) dilution.

## 4. Discussion

BCTD is one of Iran’s most damaging sugar beet diseases caused by beet curly top virus, beet curly top Iran virus, turnip curly top virus and turnip leaf roll virus. These viruses share hosts, transmission modes, and vectors, and, as a result, the occurrence of mixed infections in the field is very frequent [52]. BCTIV is one of the main causal agents. This becurtovirus is a natural recombinant virus from ancestors from the genera *Curtovirus* and *Mastrevirus*. Up to date, no studies have been conducted to study the function of any of the ORFs identified in the viral genome [27,28]. In order to better know the virus–plant interactions and the mechanisms of viral infection for this virus, we have characterized one of these ORFs, named V2. This gene is essential in viral infection and pathogenicity for many begomoviruses and some curtoviruses.

The conservation of protein domains/regions (P1, H1, or H2) present in all geminiviral V2, and involved in PTGS suppression and pathogenicity [25,39], reinforced our original supposition that BCTIV V2 would have similar activities. The results obtained in local and systemic silencing assays (Figure 2 and Figure 3) have proven that BCTIV V2: (i) is a PTGS suppressor at a local level, (ii) it produces a delay in spreading of the systemic silencing, but (iii) cannot stop the cell-to-cell movement of the silencing signal to neighboring cells. Furthermore, considering its high sequence homology with BCTV V2 protein (63% aminoacidic identity) and the conservation of the P1 domain, it is possible that, as the curtoviral protein, BCTIV V2 is suppressing PTGS by impairing RDR6/SGS3 silencing pathway, although more experiments would be needed to prove this point [16].

As BCTV V2, BCTIV V2 expression in a PVX heterologous system caused the induction of necrotic lesions in wt *N. benthamiana*, such as the ones produced during the HR, both at local and systemic levels in the wt *N. benthamiana* plants inoculated (Figure 4A,B). These results suggest that, in this PVX infection context, both proteins are avirulence factors that are directly or indirectly recognized by a resistance gene to trigger a defense response based on HR, known as effector-triggered immunity (ETI) [51]. It is remarkable that ectopic expression of BCTIV V2 from a 35S promoter did not cause any necrotic lesion in the infiltrated area in wt *N. benthamiana* leaves (similar results are observed when expressing BCTV V2) [16]. This difference could be due to an increased expression of the viral protein in this heterologous system or, maybe, as it has been suggested by some experiments performed by Aguilar and colleagues, a combination of the actions of the geminiviral suppressor being over-expressed and the one already contained in the PVX genome, P25 [53]. The differential participation of the functional domains of the BCTIV V2 protein in the HR-like response and the suppression of PTGS would need further study.

Depending on the silencing pathway steps that the viral suppressor is targeting, the protein needs to be in a specific localization inside the cell [6]. According to our subcellular localization results, BCTIV V2 is localized in the nucleus and the perinuclear region, possibly associated with RE (Figure 5A). This location has been reported for other V2 proteins that are silencing suppressors, ranging from BCTV, tomato yellow leaf curl virus (TYLCV), apple geminivirus (AGV), and mulberry mosaic dwarf-associated virus (MMDaV) [14,16,49,53]. Interestingly, in a closer examination of the nucleus, the GFP fused BCTIV V2 protein seems to be accumulating in sub-nuclear structures (Figure 5B, Appendix A). Some sub-nuclear structures, as, for example, Cajal bodies (CBs), play an important role in plant–virus interaction. It has been described that CBs are essential for systemic infection, are involved in viral systemic movement, and, in the case of DNA viruses, are involved in the suppression of TGS and viral genome methylation [54]. Interestingly, V2 from the bgomovirus TYLCV inhibits the methylation of the viral genome through its interaction with AGO4 in CBs [50,55,56]. If becurtovirus V2 also has a role in TGS, suppression will require further analysis.

The results from the infection of wt *N. benthamiana* with a BCTIV V2 mutant showed that V2 or V3 is required for systemic infection. Although the mutation in V2 generated a change in a non-conserved aminoacid of V3 (Appendix A), we cannot discharge the possibility that this change could impact the ability of the virus to infect the plant. However, the fact that the V2 mutant virus can replicate in a local infection assay, but cannot infect systematically the plant, could point to a role of V2 (or V3) in viral movement. Geminiviral V2 protein has been involved in the viral movement in other geminiviruses, such as curtoviruses, mastreviruses, and monopartite begomoviruses [1,19,20,48]. The presence of NLS in BCTIV V2 and its subcellular localization suggest that V2 could be imported to the nucleoplasm and maybe participate, along with other viral proteins, in the viral movement outside the nucleus, as it has been proved for TYLCV V2 [57].

Considering all the results, we conclude that BCTIV V2 is a functional homolog of curtoviral and begomoviral V2.

## Figures and Tables

**Figure 1 plants-11-03351-f001:**
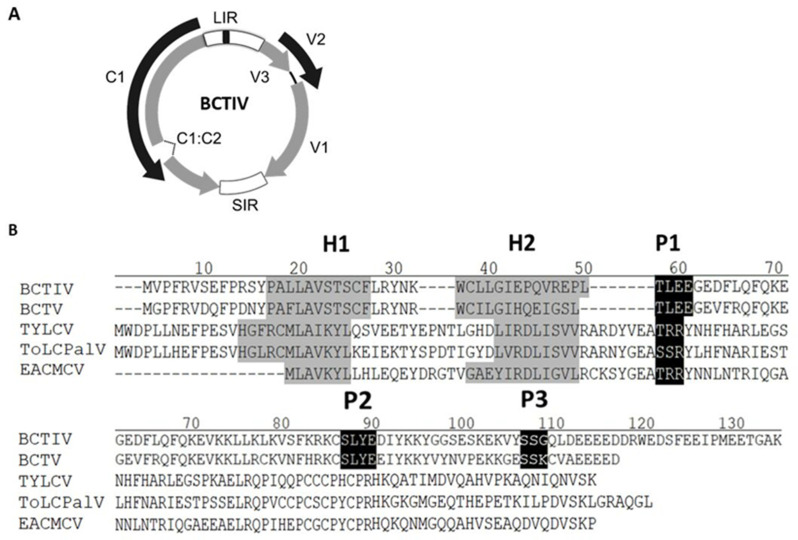
(**A**): Beet curly top Iran virus (BCTIV) genome structure. Arrows represent open reading frames (ORFs). LIR: long intergenic region; SIR: short intergenic region. Common region inside LIR is depicted in black (**B**): Alignment of the aminoacid sequences of the V2 proteins from the becurtovirus beet curly top Iran virus (BCTIV; AFK14083), the curtovirus beet curly top virus (BCTV; AAA42752.1), the begomoviruses tomato yellow leaf curl virus (TYLCV; CAA33687.1), tomato leaf curl Palampur virus (ToLCPalV; CAP03292), and East African cassava mosaic Cameron virus (EACMCV; AF112354). Gaps (-) were introduced to optimize the alignment. The positions of the predicted putative phosphorylation motifs P1 (protein kinase CK2/protein kinase C), P2 (protein kinase CK2), and P3 (protein kinase C) are depicted in white letters inside black boxes. The hydrophobic domains (H1 and H2) are shadowed in gray.

**Figure 2 plants-11-03351-f002:**
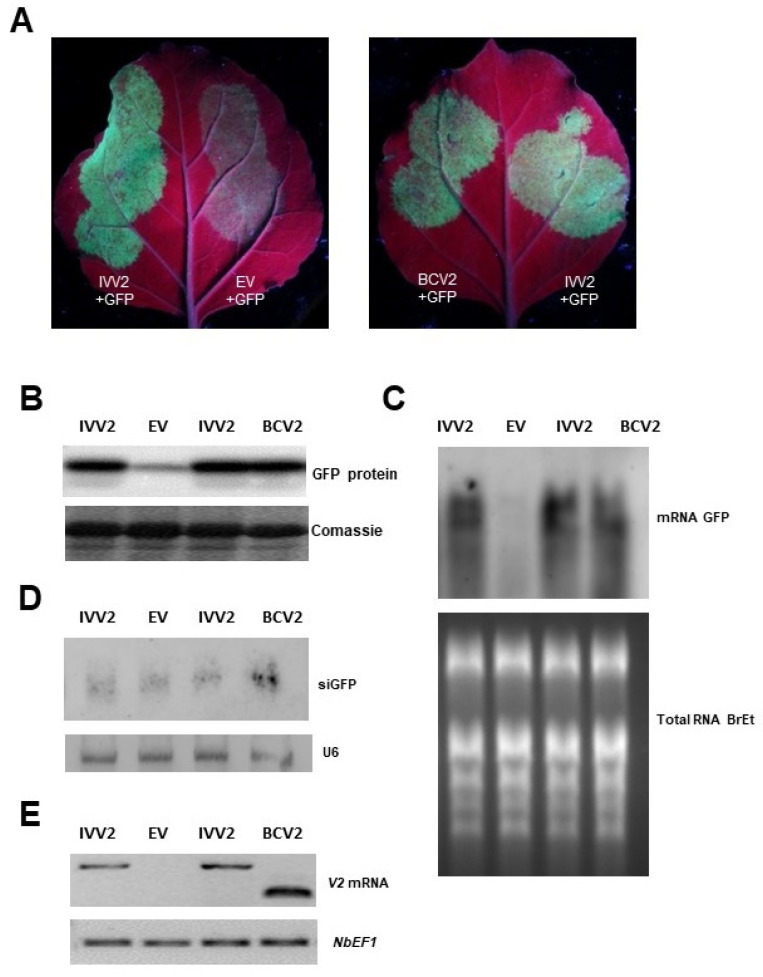
Local PTGS suppression assay in wild type *Nicotiana benthamiana*. (**A**) Leaves co-infiltrated by a mixture of two *Agrobacterium tumefaciens* cultures expressing GFP and V2 from beet curly Iran top virus (IVV2), under UV light at six days post infiltration (p.i.). V2 from beet curly top virus (BCV2) and the empty vector (EV) were used as controls in (**B**). Western blot was performed by anti-GFP antibody. Coomassie blue staining of SDS-PAGE gel is shown as a loading control. Four to six plants were agroinfiltrated per experiment. Similar results were obtained in three independent experiments. (**C**) High-molecular weight Northern blot to detect GFP mRNA accumulation in infiltrated tissues. 10 ug of total RNA was used per lane. The ethidium bromide-stained pattern is shown as a loading control. (**D**) Small RNA Northern blot analysis of GFP-derived siRNAs (siGFP) in infiltrated tissues was observed. 20 ug of total RNA per lane was observed. Detection of sn U6 is used to control for loading. (**E**) V2 viral protein transcription was confirmed by RT-PCR, and *Elongation factor alpha* (*NbE1Fa*) was used as an internal control.

**Figure 3 plants-11-03351-f003:**
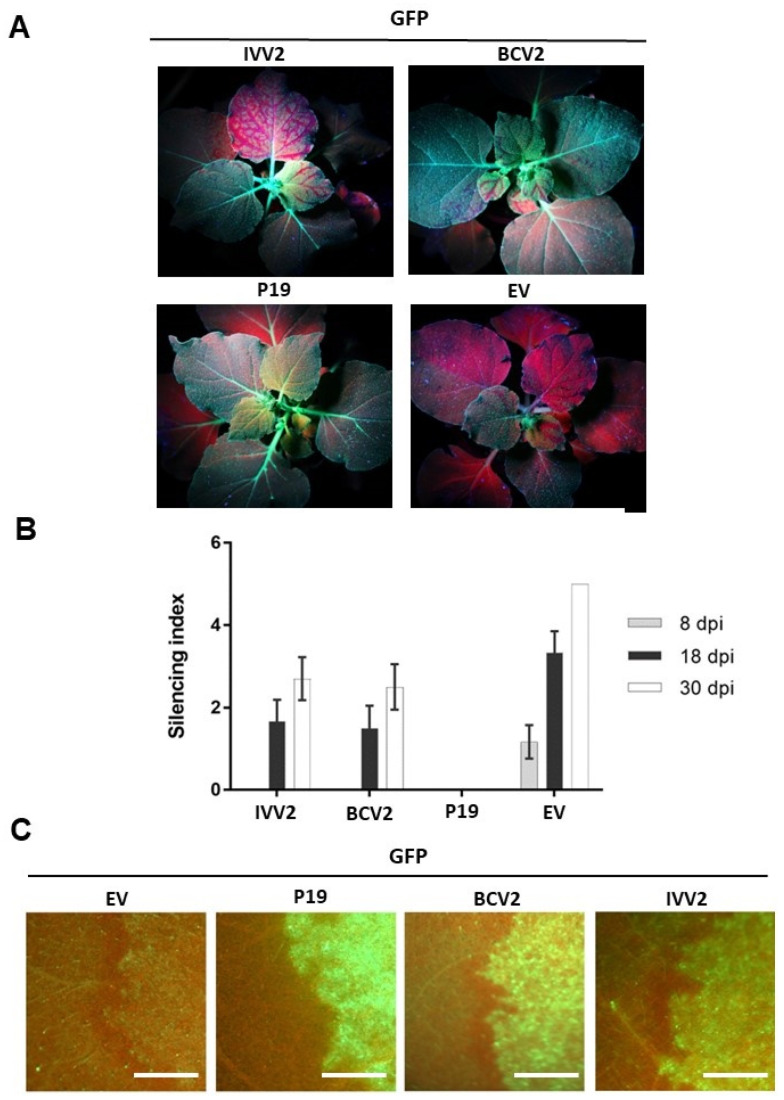
Effect of V2 from beet curly top Iran virus on long distance and short range spread of RNA silencing in 16c *Nicotiana benthamiana* plants. (**A**) 16c plants were agroinfiltrated with two *A. tumefaciens* cultures expressing GFP and V2 from beet curly top Iran virus (IVV2). V2 from beet curly top virus (BCV2), P19, and the empty vector (EV) were used as controls. Agroinfiltrated 16 c plants were observed under UV light at 18 days post inoculation (dpi). (**B**) Levels of systemic silencing (silencing index) of these plants at 8-, 18-, and 30-days post infiltration (dpi). Silencing index ranges from 0 (no silenced leaves) to 5 (plant with all the leaves silenced). Values correspond to the average of six plants. Bars represent standard deviation. (**C**) GFP expression in the cells surrounding the agroinfiltrated area at 6 dpi. Barr: 2 mm. Four to six plants were agroinfiltrated per experiment. Similar results were obtained in three independent experiments.

**Figure 4 plants-11-03351-f004:**
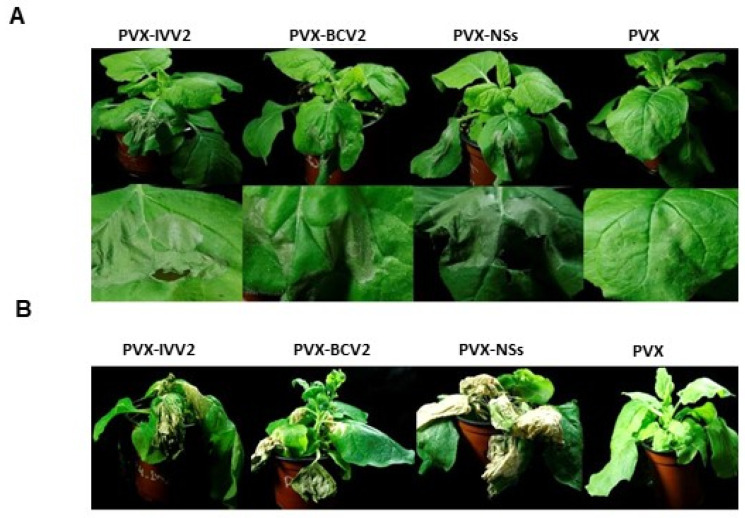
Local and systemic symptoms induced by *Potato virus* X (PVX) expression of V2 from beet curly top Iran virus (PVX-IVV2) in wt *Nicotiana benthamiana* plants. Empty PVX vector (PVX) was used as negative control. Recombinant PVX viruses expressing either the nonstructural protein (NSs) from tomato spotted wilt virus (PVX-NSs) or the V2 protein from beet curly top virus (PVX-BCV2) were used as positive controls. (**A**) Local symptoms induced in the infiltrated area at 6 dpi. (**B**) Representative plants showing systemic symptoms at 16 dpi. Four to six plants were agroinfiltrated per experiment. Similar results were obtained in three independent experiments.

**Figure 5 plants-11-03351-f005:**
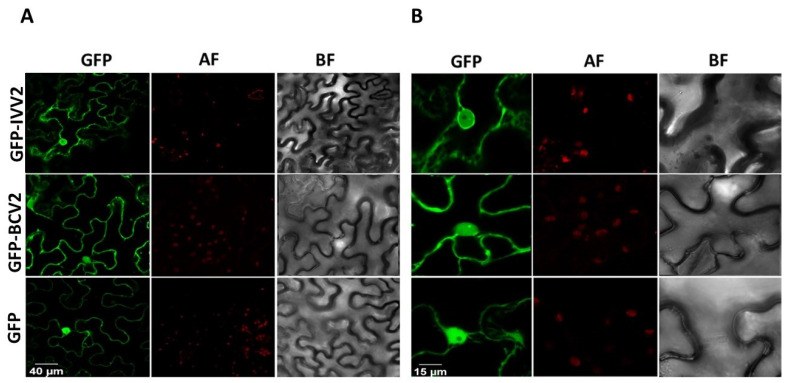
Subcellular localization of V2 from beet curly top Iran virus fused to GFP in epidermal cells of wild type *Nicotiana benthamiana*. (**A**) Leaves were agroinfiltrated with a construct expressing the GFP (GFP), GFP-V2 fusion protein from beet curly top Iran virus (GFP-IVV2), or the GFP-V2 fusion protein from beet curly top virus (GFP-BCV2). Samples were observed under the confocal microscope at 36 h post infection. (**B**) Close up confocal images of the observed areas. GFP fluorescence (GFP), autofluorescence (AF), and the bright field channel (BF) are shown.

**Figure 6 plants-11-03351-f006:**
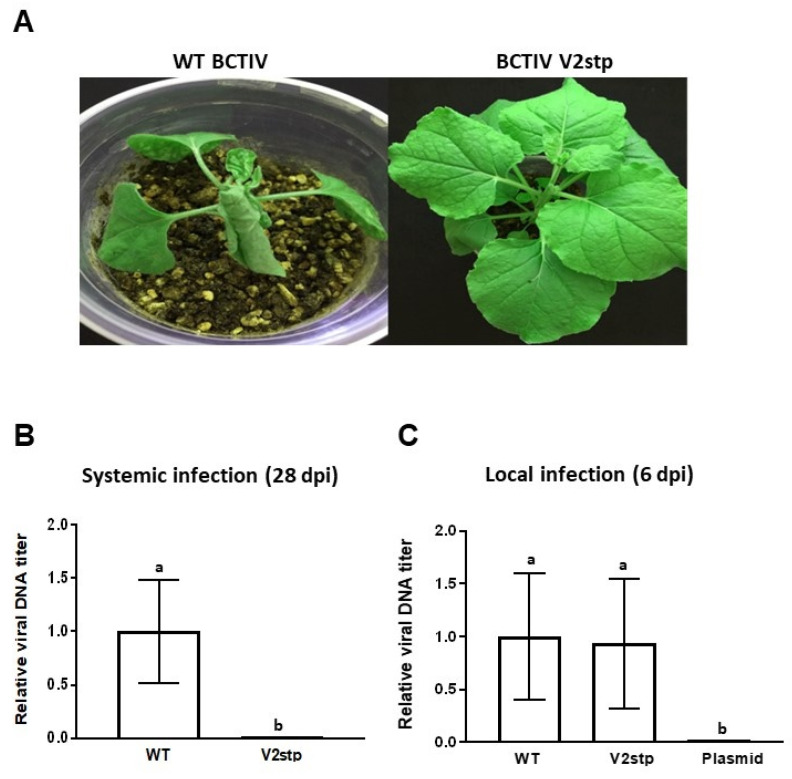
Infection of wild type *Nicotiana benthamaiana* plants with wild type beet curly top Iran virus and the V2 stop mutant virus. Plants were agroinoculated with wild-type (wt BCTIV) or V2 mutant BCTIV (BCTIV V2stp) infectious clones. (**A**) Representative plants showing BCTIV symptoms at 28 dpi. Relative viral DNA accumulation in apical leaves (systemic infection) at 28 dpi (**B**) or in the infiltrated leaves (local infection) at 6 dpi (**C**). BCTIV accumulation was measured by qPCR after *DpnI* treatment to remove bacterial DNA (6 dpi). The same *DpnI* treatment was applied on bacterial plasmids containing the BCTIV infectious clone as a control (Plasmid). DNA levels were normalized to *Elongation factor alpha* (*NbE1Fa*) and are presented as the relative amount of virus compared with the amount found in wild-type BCTIV samples (set to 1). Bars represent mean values ± standard error (SE) for six pools of two leaves. Mean values marked with different letters (a or b) indicate results significantly different from each other, as established by Student’s *t*-test α < 0.001) virus titer.

## Data Availability

Not applicable.

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
