# Peer review of "Functional Analysis of V2 Protein of Beet Curly Top Iran Virus"

_plants, 2022, doi:10.3390/plants11233351_

Round 1
Reviewer 1 Report
In this paoer the authors study the beet curly top Iran virus (BCTIV) of beets, a new member of Becurtoviruses. genus type species., that shares a limited sequence homology with curtoviruses. To address if a function of BCTIV V2 is a local silencing suppressor delaying the spread of silencing in Nicotiana benthamiana they show the same subcellular localization as curtoviral V2 in the nucleus and perinuclear region. Similarly, its ectopic expression from a PVX-vector induced necrotic lesions in N. benthamiana during the HR. Whereas the infection with a V2 BCTIV mutant confirmed the role of V2 in systemic infection. Therefore, the authors conclude that BCTIV V2 is functional homologue of curtoviral V2.
This is an important work that should be published. Introduction is well written and informative. Likewise are Materials and Methods. The Results part is well documented, and Discussion addresses most important aspects of the obtained results.
One thing I note is that a mutation leading to generation of an early stop codon in V2 ORF also replaces the Arg to Leu in the overlapping V3 gene, which might lead to unpredictable side effects. This aspect should be clearly discussed in the Discussion part.
Author Response
In this paper the authors study the beet curly top Iran virus (BCTIV) of beets, a new member of Becurtoviruses. genus type species., that shares a limited sequence homology with curtoviruses. To address if a function of BCTIV V2 is a local silencing suppressor delaying the spread of silencing in Nicotiana benthamiana they show the same subcellular localization as curtoviral V2 in the nucleus and perinuclear region. Similarly, its ectopic expression from a PVX-vector induced necrotic lesions in N. benthamiana during the HR. Whereas the infection with a V2 BCTIV mutant confirmed the role of V2 in systemic infection. Therefore, the authors conclude that BCTIV V2 is functional homologue of curtoviral V2.
This is an important work that should be published. Introduction is well written and informative. Likewise are Materials and Methods. The Results part is well documented, and Discussion addresses most important aspects of the obtained results.
One thing I note is that a mutation leading to generation of an early stop codon in V2 ORF also replaces the Arg to Leu in the overlapping V3 gene, which might lead to unpredictable side effects. This aspect should be clearly discussed in the Discussion part.
We thank the reviewer for his/her comments. We have rewritten this part in the discussion and in the results sections to make it clear that we cannot rule out that the phenotype we observed could be also due to V3.
Reviewer 2 Report
Dear authors
I read the Functional Analysis of V2 Protein of Beet Curly Top Iran Virus article. It is an interesting subject.
General comments
Please answer my questions:
References are not suitable for the MDPI format. Please revise them.
What is the novelty of this study? (It is not clear to me)
Author Response
Dear authors
I read the Functional Analysis of V2 Protein of Beet Curly Top Iran Virus article. It is an interesting subject.
General comments
Please answer my questions:
We thank the reviewer for her/his comments and suggestions.
References are not suitable for the MDPI format. Please revise them.
We have revised the references and changed the format accordingly in the manuscript.
What is the novelty of this study? (It is not clear to me).
BCTIV is a relevant pathogen being one of the main causal agents of the beet curly top disease in Iran, disease that has a great impact on agriculture. Although some of BCTIV proteins have a similar sequence than the positional homologues in curtoviruses, there is not information available on the function of any of these viral proteins. In our study we established for the first time, that BCTIV V2 is a PTGS silencing suppressor and plays a crucial role in viral pathogenicity and movement.
Reviewer 3 Report
Review comments for Plants (ISSN 2223-7747)
The manuscript showed results of functional analyses of the geminivirus beet curly top Iran virus (BCTIV) V2 protein and concluded conclude that the BCTIV V2 is a functional homolog of curtovirus and begomovirus V2 proteins. While showing some evidence of BCTIV V2’s involvements in working as a viral silencing suppressor, in viral long-distance movement or systemic infection, some essential experiments and controls are missing in the manuscript. The following are the comments/suggestions for this manuscript:
1. Most of the genus, species names of plants and virus were not written in the correct ways (most are not italicized). For example: line 11, “Becurtovirus genus….” The Becurtovirus is not italicized. Most of Nicotiana benthamiana, Agrobacterium are not italicized. Additionally, the virus names are not supposed to be capitalized in the middle of sentences. Some are correct, most are in correct in the manuscript. No consistency.
2. P1-Line 37, “…viral small interfering RNAs (vsiRNAs)” should be “virus-derived small interfering RNAs”.
3. P2-Line 40, “2.2 Bioinformatic analysis”; Should be “sequence analysis”. There’s no big data involved in this study. The word bioinformatic is not adequate here.
4. P2-Line 47, “Generation of binary vectors” is not a good subtitle here. The authors didn’t generate binary vectors but constructed virus infectious clones or protein expression clones in binary vectors.
5. P3-Line 46, the authors didn’t describe how they designed their “semi-“quantitative RT-PCR, the conditions of those RT-PCR reactions.
6. P4-Line 18, “bioinformatic analysis” should be sequence analysis or sequence alignment analysis.
7. P4-Line 20-21, “begomoviral ones” can just be begomoviruses.
8. P5 Figure 2, The authors should show the accumulation level differences of the small RNAs and transcripts derived from GFP in each treatment to prove that V2 proteins are directly involved in PTGS pathways.
9. P7 Figure 3A, Image quality/resolution is too low for all the images presented in the figure 3. The experiments also lack positive controls. P6-Line 19-20, “…IVV2 or BCV2, suggesting neither of these proteins is able to suppress the spreading of the GFP silencing.” So, there's no proof that the method that's done here could be used for suppression of systemic silencing.
10. P7 Figure 3B, Addition to the low-resolution issue, the figure 3B is lacking scale bars in all the images.
11. P7-Line 12, “…surrounding the infiltrated tissues…” How were the infiltrated tissues were /marked? How did authors know that the free GFP didn’t move cell-to-cell?
12. P7-Line 16, “This red ring…” The image resolution is too low. There's no clear "red ring" in the figure 3B.
13. P8 Figure 4, Image resolutions in the figure 4 are too low. Figure 4B: Although BCV2 was used as a positive control, PVX-BCV2 is not showing strong systemic necrosis as others.
14. P9-Line 5, “detected” should be observed.
15. P9-Line 6, GFP-BCV2 has no obvious difference compared to the free GFP signals. The signal in the GFP image in figure 5A is weak unlike other regular free GFP expression.
16. P10-Line 17-18, 21, Figure 6: Controls that the qPCR reactions would not detect the infiltrated plasmids are needed. The data showing here is not sufficient to prove that BCTIV V2 is not required for replication.
17. P11-Line 24-25, The results can only prove that the V2 genes are avirulence factors during PVX infection, but not for their respective viruses. The authors should provide such evidence using BCTV or BCTIV infectious clones to prove their claims about the V2 proteins being avirulence factors.
18. P11-Line 43-44, “…remeble Cajal bodies” There is no markers for Cajal bodies in the Figure 5B and supplementary figure 2).
19. The numbering of the supplementary figures needs to be corrected. There are two supplementary figure 2s.

Author Response
The manuscript showed results of functional analyses of the geminivirus beet curly top Iran virus (BCTIV) V2 protein and concluded conclude that the BCTIV V2 is a functional homolog of curtovirus and begomovirus V2 proteins. While showing some evidence of BCTIV V2’s involvements in working as a viral silencing suppressor, in viral long-distance movement or systemic infection, some essential experiments and controls are missing in the manuscript. The following are the comments/suggestions for this manuscript:
We thank the reviewer for his/her comments and suggestions that have helped us to improved the quality of the manuscript.
- Most of the genus, species names of plants and virus were not written in the correct ways (most are not italicized). For example: line 11, “Becurtovirus genus….” The Becurtovirus is not italicized. Most of Nicotiana benthamiana, Agrobacteriumare not italicized. Additionally, the virus names are not supposed to be capitalized in the middle of sentences. Some are correct, most are in correct in the manuscript. No consistency. We have revised the names and corrected the mistakes that we have detected.
- P1-Line 37, “…viral small interfering RNAs (vsiRNAs)” should be “virus-derived small interfering RNAs”. We have changed it accordingly.
- P2-Line 40, “2.2 Bioinformatic analysis”; Should be “sequence analysis”. There’s no big data involved in this study. The word bioinformatic is not adequate here. We have changed it accordingly.
- P2-Line 47, “Generation of binary vectors” is not a good subtitle here. The authors didn’t generate binary vectors but constructed virus infectious clones or protein expression clones in binary vectors. We have changed it accordingly.
- P3-Line 46, the authors didn’t describe how they designed their “semi-“quantitative RT-PCR, the conditions of those RT-PCR reactions. We have described the conditions with more details.
- P4-Line 18, “bioinformatic analysis” should be sequence analysis or sequence alignment analysis. We have changed it accordingly.
- P4-Line 20-21, “begomoviral ones” can just be begomoviruses. We have changed it accordingly.
- P5 Figure 2, The authors should show the accumulation level differences of the small RNAs and transcripts derived from GFP in each treatment to prove that V2 proteins are directly involved in PTGS pathways. We have performed a northern blot to detect siRNA and mRNA GFP accumulation in agroinfiltrated tissues. We have modified the figure 2 accordingly to include this new analysis. We detected a clear correlation between the levels of mRNA and protein, with a marked decrease in the tissues co-infiltrated with GFP and the empty vector compared to the ones co-infiltrated with GFP and BCTIV or BCTV V2 proteins. However, there were no significant differences between GFP siRNA accumulation in the different experiments.
- P7 Figure 3A, Image quality/resolution is too low for all the images presented in the figure 3. The images with more resolution can be found in the supplementary materials. The experiments also lack positive controls. P6-Line 19-20, “…IVV2 or BCV2, suggesting neither of these proteins is able to suppress the spreading of the GFP silencing.” So, there's no proof that the method that's done here could be used for suppression of systemic silencing. Our positive control is co-infiltration with P19. As it is shown in our results, this suppressor stops completely the spreading of GFP silencing in this system (P6. L25: “as expected in plants infiltrated with P19 no systemic silencing was observed”).
- P7 Figure 3B, Addition to the low-resolution issue, the figure 3B is lacking scale bars in all the images. We have changed it accordingly.
- P7-Line 12, “…surrounding the infiltrated tissues…” How were the infiltrated tissues were /marked? How did authors know that the free GFP didn’t move cell-to-cell? The infiltrated area is marked just after agroinfiltration and we could check that the GFP fluorescence remained inside the infiltrated area. In Himber et al 2003 they co-infiltrated GUS and GFP and described that transgene expression remained inside the infiltrated area, but the silencing signal was able to exit this area (10-15 cells around) (Himber C, Dunoyer P, Moissiard G, Ritzenthaler C, Voinnet O. Transitivity-dependent and -independent cell-to-cell movement of RNA silencing. EMBO J. 2003 Sep 1;22(17):4523-33. doi: 10.1093/emboj/cdg431. PMID: 12941703; PMCID: PMC202373.)
- P7-Line 16, “This red ring…” The image resolution is too low. There's no clear "red ring" in the figure 3B. We have changed the figure adding images with more resolution and contrast to make easier to appreciate the red ring. The images with more resolution can be found in the supplementary materials.
- P8 Figure 4, Image resolutions in the figure 4 are too low. Figure 4B: Although BCV2 was used as a positive control, PVX-BCV2 is not showing strong systemic necrosis as others. We have modified the figure according to the requirements. We have found more representative pictures of the systemic necrosis where the differences between the necrosis caused by BCV2 and IVV2 are not so marked. The images with more resolution can be found in the supplementary materials
- P9-Line 5, “detected” should be observed. We have changed it accordingly.
- P9-Line 6, GFP-BCV2 has no obvious difference compared to the free GFP signals. The signal in the GFP image in figure 5A is weak unlike other regular free GFP expression. We have modified the figure to enhance the contrast and resolution (The images with more resolution can be found in the supplementary materials). In the GFP-BCV2 infiltrated tissues, as in GFP-IVV2, there are also punctuate bodies that are not present when free GFP is expressed. As it is shown in the western blot (Supp. Fig. 2), most of the GFP present in GFP-BCV2 infiltrated tissues is present as a fusion protein, indicating that the observed localization is mainly due to BCV2 and not to free GFP.
- P10-Line 17-18, 21, Figure 6: Controls that the qPCR reactions would not detect the infiltrated plasmids are needed. The data showing here is not sufficient to prove that BCTIV V2 is not required for replication. We have modified the graphic in the figure and included the data from a control (a sample with bacterial plasmid containing the infectious BCTIV clone treated with DpnI). Since amplification of the viral DNA was only obtained in the samples from local infection but not the control ones, we can conclude that we are detecting replicating DNA from the virus and not any residual bacterial DNA. We also have included the description of this control in the text (P3, L48-49) and in the figure 6 legend.
- P11-Line 24-25, The results can only prove that the V2 genes are avirulence factors during PVX infection, but not for their respective viruses. The authors should provide such evidence using BCTV or BCTIV infectious clones to prove their claims about the V2 proteins being avirulence factors. From our results we conclude that V2 is a virulence factor in a PVX infection scenario, we have added a sentence to make it clearer
(P12, L25-30 “As BCTV V2, BCTIV V2 expression in a PVX heterologous system caused the in-duction of necrotic lesions in wt N. benthamiana like the ones produced during the HR, both at local and systemic levels in the wt N. benthamiana plants inoculated. (Fig. 4A, B). These results suggest that in this PVX-infection context, both proteins are avirulence factors that are ….”)
The fact that BCTV infections don’t cause necrosis could indicate that V2 levels are not enough to cause this or that there are other viral proteins counteracting this effect. To determine which are the causes we would need to do additional experiments. A sentence has been now included in the discussion.
- P11-Line 43-44, “…remeble Cajal bodies” There is no markers for Cajal bodies in the Figure 5B and supplementary figure 2). We have removed this comparation from the paragraph. It is true that without colocalization studies with a Cajal body marker, we cannot affirm that the observed subnuclear bodies are Cajal bodies. On the other hand, we only would like to discuss a possible role of the observed structures in suppression of RNA silencing, given that this role has been described for CBs and the localization of begomovirus V2 protein in these structures has been shown to be important for its function
- The numbering of the supplementary figures needs to be corrected. There are two supplementary figure 2s. We have changed it accordingly.
Reviewer 4 Report
The manuscript entitled “Functional Analysis of V2 Protein of Beet Curly Top Iran Virus” by Bahari and colleagues highlights for the first time the role of V2 of BCTIV as a local suppressor of RNA silencing. The authors performed a series of experiments to identify and clarify the role of V2 in the RNA silencing machinery and in general I find the methodology and results satisfying and adequate to prove their hypothesis. The major drawback of the manuscript lays in the part of the V2 mutant in the infectious clone that also resulted in the mutation of V3 (as they partially overlap). I recommend the authors to read carefully this part and revise it accordingly to the commends below. Also, I quote my revision on the manuscript and I believe that its consideration with improve the quality of the manuscript and reward the authors for all this interesting and thorough work. Therefore, my recommendation is acceptance with revision.
Page 1-2: Introduction
I believe that the introduction could be reshaped and perhaps a few sentences-paragraphs may be omitted. You start with the family Geminiviridae, then you describe the RNA silencing defense mechanism and then you go back to geminiviruses, their V2 proteins, the V2 protein of BCTV (which is not the virus you studied) and then to BCTIV. I believe this structure is a bit chaotic and may be shaped to a simpler one such as the following: you may start with describing the RNA silencing mechanism, then with a couple of phrases (max) on geminiviruses and then to BCTIV and its encoded proteins. All the other information may be implemented in the discussion part.
Page 2.
L45. “The primer sequences used in …”
L32. The term ‘manipulation’ is a bit too general to describe an experimental process, perhaps you should replace it with transformation and be a little more specific when needed.
L25. Since in your experiments you used WT and 16c N. benthamiana plants, I would recommend to add this clarification every time you refer to N. benthamiana (unless you are talking about N. benthamiana in general).
Page 3.
L13. Replace ‘infective’ with ‘infectious’. Also replace this term in results and discussion if/when used.
L38. “Nucleic acid extraction…”. Also is it total nucleic acid or RNA?
Page 4
L7. I am not sure if I agree with the term ‘strong’ for the silencing suppressor activity. You also use this word in the next page in L.15, 21 and 22. You may use the word efficient if you have experimental data that support this (efficient compared with ..?) or you may commend on its lasting suppression activity if again you have any data. But based on your results, you should omit the word ‘strong’ from the title. Regarding the next page L15: patches co-infiltrated with 35:GFP and IVV2 displayed high-intensity of GFP fluorescence compared to the positive.
L8-21. I believe this paragraph need another subtitle. The in silico analysis you performed does not indicate that BCTIV-V2 is a strong local PTGS suppressor, your next paragraph describes that.
Page 5.
L11. Did all the other constructs (empty vector, BCV2 etc) were on plasmids with 35S? If yes, you should use the same format as 35S:GFP for the rest of them: 35S:BCTIVV2, 35S:EV, 35S:BCV2 etc….
L15. Co-infilrated with 35S:GFP and BCV2
L16. “… empty vector, the intensity of GFP fluorescence...”
L18. Delete ‘As expected’ as in this paragraph you are still trying to make a case for your protein acting as a putative VSR. It is not something well-established yet.
Figure 2. Which is the difference between IVV2 (1) and IVV2 (2)? Describe it in the legend. Also delete ‘(N. benthamiana)’
Since cell-to-cell movement of the silencing signal precedes the systemic movement, you should first describe the results from 3.3 and then from 3.2. This would also change the array you put your figures in Figure 3.
Page 6.
L20. Perhaps replace ’suppress the spreading of GFP silencing’ with ‘interfere with the systemic movement of RNA silencing’.
L24. This delay in the spread of the systemic movement of RNA silencing is also depicted in Supp. Figure 2 (check the numbering in Supplementaries). I believe that Suppl. Fig. 2 can move to Figure 3 (perhaps 3C) because it consists an important part of the result.
Page 7.
L19. Delete ‘as its Curtoviral counterpart”
Subtitles 3.1-3.6. Replace “V2 of BCTIV’ by “BCTIV V2”
Page 8.
L.14. ‘…necrosis, the symptoms are more severe for..’
Page 9.
L19-21. I find this part problematic. If your site directed mutagenesis on V2 affected also V3, how are you sure that the inability of systemic infection is due to V2 and not to the mutation of V3 as well (or both)? You mention this problem in page 11 L52-53.. but the discussion in this part is weak. Based on your experiments, you cannot exclude the role of V3 in the systemic infection.
Regarding the next sentence ‘The fact that no differences with the wild-type viruses are observed in local infection/viral replication pointed to a role of V2 in viral movement’ I cannot find why this lack of differences in the first case leads to the conclusion for the second case. This paragraph needs to be reshaped completely and conclusions should be drawn very carefully.
L24 ‘remained asymptomatic’ this implies that you had an infection but the virus remained latent for some reason. And if I am not mistaken, you did not have an (systemic) infection at all. So perhaps you should rephrase this to ‘did not show any symptoms..’ as you correctly state in the next sentence.
DISCUSSION
Page 11.
L2. You mention “these viruses” as if you have stated the virus species in the previous sentence, whereas you only mentioned that the viruses implicated in BCTB belong to three genera. I would recommend to add the virus species in the first sentence of this paragraph and then you can proceed with BCTIV. Or, you may start the discussion part directly with BCTIV and how the virus is involved in BCTD.
L13. I am not sure about the conclusion “Given the conservation of protein domains/regions (P1, H1, or H2) present in all geminiviral V2 and involved in PTGS suppression and pathogenicity, it was expected that BCTIV V2 would have similar activities”. I believe that a better way to present you case is to explain why you chose to test V2 as a candidate VSR, and one of the reasons in that the same protein has a similar function to geminivirus cognate (as you correctly explain). The fact that you detected conserved domains/regions reinforces that case but it is not something that was expected from the beginning.
L15 delete ‘strong’
L19-20. Is 63% nt or aa sequence similarity? There are cases in which VSR proteins, with ‘high’ sequence similarity, play different roles in RNA silencing pathway. Here you mention a probable role of BCTIV V2 but you did not perform any experiments
Supplementary Figure 3 (or 4)
The citation of this figure may also be added in the 3.6 section in L21.
Page 11.
L52. BCTIV V2 mutant..
General comments
-Some terms/phrases/words need to be replaced by more appropriate/accurate/scientifically-sound ones. For example:
P5-L15. Was still strong
P6-L22. In the extension of the silenced tissues
P6-L24. In the silencing spreading
P10-L16: BCTIV V2 stop could not infect plants systemically (the word stop, in other parts you refer it as mutant which I believe is more appropriate)
P11-L17. Cannot avoid
P12-L2. Wild-type viruses
-Lack of ‘.’ Or too many “..”In the end of phrases/sentences (for example P10-L11, P12-L10, P2-L39, P2-L3, P12-L20 etc.)
-Since in all your agroinfiltation assays you co-infiltrate 35S-GFP with 35S:-protein, you cannot say that ‘the patches infiltrated with BCV2’ or ‘p19’ alone. I have seen this mistake in many parts of mats and methods and results so these parts need corrections.
Author Response
The manuscript entitled “Functional Analysis of V2 Protein of Beet Curly Top Iran Virus” by Bahari and colleagues highlights for the first time the role of V2 of BCTIV as a local suppressor of RNA silencing. The authors performed a series of experiments to identify and clarify the role of V2 in the RNA silencing machinery and in general I find the methodology and results satisfying and adequate to prove their hypothesis. The major drawback of the manuscript lays in the part of the V2 mutant in the infectious clone that also resulted in the mutation of V3 (as they partially overlap). I recommend the authors to read carefully this part and revise it accordingly to the commends below. Also, I quote my revision on the manuscript and I believe that its consideration with improve the quality of the manuscript and reward the authors for all this interesting and thorough work. Therefore, my recommendation is acceptance with revision.
We thank the reviewer for her/his suggestions that have helped us to improve the manuscript quality.
Page 1-2: Introduction
I believe that the introduction could be reshaped and perhaps a few sentences-paragraphs may be omitted. You start with the family Geminiviridae, then you describe the RNA silencing defense mechanism and then you go back to geminiviruses, their V2 proteins, the V2 protein of BCTV (which is not the virus you studied) and then to BCTIV. I believe this structure is a bit chaotic and may be shaped to a simpler one such as the following: you may start with describing the RNA silencing mechanism, then with a couple of phrases (max) on geminiviruses and then to BCTIV and its encoded proteins. All the other information may be implemented in the discussion part.
We have rewritten and reorganized the introduction following the suggestions of the reviewer.
Page 2.
L45. “The primer sequences used in …” We have changed it accordingly.
L32. The term ‘manipulation’ is a bit too general to describe an experimental process, perhaps you should replace it with transformation and be a little more specific when needed. We have rewrite it accordingly.
L25. Since in your experiments you used WT and 16c N. benthamiana plants, I would recommend to add this clarification every time you refer to N. benthamiana (unless you are talking about N. benthamiana in general). We have added the terms 16c or wt to N. benthamiana to clarify the text.
Page 3.
L13. Replace ‘infective’ with ‘infectious’. Also replace this term in results and discussion if/when used. We have changed it accordingly.
L38. “Nucleic acid extraction…”. Also is it total nucleic acid or RNA? We have rewritten this part adding more details.
Page 4
L7. I am not sure if I agree with the term ‘strong’ for the silencing suppressor activity. You also use this word in the next page in L.15, 21 and 22. You may use the word efficient if you have experimental data that support this (efficient compared with ..?) or you may commend on its lasting suppression activity if again you have any data. But based on your results, you should omit the word ‘strong’ from the title. Regarding the next page L15: patches co-infiltrated with 35:GFP and IVV2 displayed high-intensity of GFP fluorescence compared to the positive. We have changed it accordingly.
L8-21. I believe this paragraph need another subtitle. The in silico analysis you performed does not indicate that BCTIV-V2 is a strong local PTGS suppressor, your next paragraph describes that. We have added a new section for this paragraph entitled: “ 3.1. BCTIV V2 aminoacidic sequence analysis” and we have changed the title for the next one to “3.2. BCTIV V2 is a local PTGS suppressor”
Page 5.
L11. Did all the other constructs (empty vector, BCV2 etc) were on plasmids with 35S? If yes, you should use the same format as 35S:GFP for the rest of them: 35S:BCTIVV2, 35S:EV, 35S:BCV2 etc…. We have removed the 35S: from the GFP given that all the plasmids contain a 35S promoter.
L15. Co-infiltrated with 35S:GFP and BCV2. We have changed it accordingly
L16. “… empty vector, the intensity of GFP fluorescence...” We have changed it accordingly
L18. Delete ‘As expected’ as in this paragraph you are still trying to make a case for your protein acting as a putative VSR. It is not something well-established yet. It has been deleted.
Figure 2. Which is the difference between IVV2 (1) and IVV2 (2)? Describe it in the legend. Also delete ‘(N. benthamiana)’ The numbers corresponded to two pools of leaves infiltrated with IVV2. In order to make easier the figure we have removed the numbers, given that both samples are infiltrated with BCTIV V2 (IVV2).
Since cell-to-cell movement of the silencing signal precedes the systemic movement, you should first describe the results from 3.3 and then from 3.2. This would also change the array you put your figures in Figure 3. We thank the suggestion of the reviewer, however we’d rather like to maintain the order as it is, since the delay in systemic silencing is the most relevant phenotype, and, as we have observed, this delay seems to be independent of the formation of the ring.
Page 6.
L20. Perhaps replace ’suppress the spreading of GFP silencing’ with ‘interfere with the systemic movement of RNA silencing’. We have modified the sentences to reflect that there are no total suppression of the GFP silencing, but there is a delay of this silencing extension, so they are able to interfere with the systemic movement, but not stop it completely as P19 does.
L24. This delay in the spread of the systemic movement of RNA silencing is also depicted in Supp. Figure 2 (check the numbering in Supplementaries). I believe that Suppl. Fig. 2 can move to Figure 3 (perhaps 3C) because it consists an important part of the result. We have moved the silencing index figure from supplementary to Figure 3 as the reviewer suggested.
Page 7.
L19. Delete ‘as its Curtoviral counterpart” We have changed it accordingly
Subtitles 3.1-3.6. Replace “V2 of BCTIV’ by “BCTIV V2”. We have changed it accordingly
Page 8.
L.14. ‘…necrosis, the symptoms are more severe for.’ We have changed it accordingly
Page 9.
L19-21. I find this part problematic. If your site directed mutagenesis on V2 affected also V3, how are you sure that the inability of systemic infection is due to V2 and not to the mutation of V3 as well (or both)? You mention this problem in page 11 L52-53.. but the discussion in this part is weak. Based on your experiments, you cannot exclude the role of V3 in the systemic infection.
Regarding the next sentence ‘The fact that no differences with the wild-type viruses are observed in local infection/viral replication pointed to a role of V2 in viral movement’ I cannot find why this lack of differences in the first case leads to the conclusion for the second case. This paragraph needs to be reshaped completely and conclusions should be drawn very carefully. We have rewritten this paragraph accordingly.
L24 ‘remained asymptomatic’ this implies that you had an infection but the virus remained latent for some reason. And if I am not mistaken, you did not have an (systemic) infection at all. So perhaps you should rephrase this to ‘did not show any symptoms..’ as you correctly state in the next sentence. We have changed it accordingly
DISCUSSION
Page 11.
L2. You mention “these viruses” as if you have stated the virus species in the previous sentence, whereas you only mentioned that the viruses implicated in BCTB belong to three genera. I would recommend to add the virus species in the first sentence of this paragraph and then you can proceed with BCTIV. Or, you may start the discussion part directly with BCTIV and how the virus is involved in BCTD. We added the virus species as recommended.
L13. I am not sure about the conclusion “Given the conservation of protein domains/regions (P1, H1, or H2) present in all geminiviral V2 and involved in PTGS suppression and pathogenicity, it was expected that BCTIV V2 would have similar activities”. I believe that a better way to present you case is to explain why you chose to test V2 as a candidate VSR, and one of the reasons in that the same protein has a similar function to geminivirus cognate (as you correctly explain). The fact that you detected conserved domains/regions reinforces that case but it is not something that was expected from the beginning. We have changed it accordingly.
L15 delete ‘strong’ We have changed it accordingly.
L19-20. Is 63% nt or aa sequence similarity? There are cases in which VSR proteins, with ‘high’ sequence similarity, play different roles in RNA silencing pathway. Here you mention a probable role of BCTIV V2 but you did not perform any experiments We have modified the text to mention here that this 63% in aminoacidic sequence together with the conservation of the domains suggest this possible role, although we also mention that to affirm this, we would need to do more experiments.
Supplementary Figure 3 (or 4) The citation of this figure may also be added in the 3.6 section in L21. We have added it.
Page 11.
L52. BCTIV V2 mutant.. We have changed it accordingly.
General comments
-Some terms/phrases/words need to be replaced by more appropriate/accurate/scientifically-sound ones. For example:
P5-L15. Was still strong We have changed it accordingly.
P6-L22. In the extension of the silenced tissues and P6-L24. In the silencing spreading. We have modified the text accordingly.
P10-L16: BCTIV V2 stop could not infect plants systemically (the word stop, in other parts you refer it as mutant which I believe is more appropriate). We have changed it accordingly.
P11-L17. Cannot avoid. We have changed it accordingly.
P12-L2. Wild-type viruses. We have removed this sentence and replaced by another one.
-Lack of ‘.’ Or too many “..”In the end of phrases/sentences (for example P10-L11, P12-L10, P2-L39, P2-L3, P12-L20 etc.) We have checked the text and added all the stop punctuations.
-Since in all your agroinfiltation assays you co-infiltrate 35S-GFP with 35S:-protein, you cannot say that ‘the patches infiltrated with BCV2’ or ‘p19’ alone. I have seen this mistake in many parts of mats and methods and results so these parts need corrections. We have revised the text and changed it accordingly.
Round 2
Reviewer 3 Report
The suggested experiments were added to better support the conclusions with all the results. Here are a couple more comments/suggestions for the manuscript:
1. Page 4 line 1: I don't think people usually use the phrase "high-weight" RNA northern blots. I would suggest changing it to high molecular weight northern blots.
2. Page 8 line 38: "... the intensity the symptoms are more severe for ..." Does this mean "... the intensity of the symptoms is more severe for ..." ?
Author Response
We thank again the reviewer for his/her suggestions, and we are very glad that the additional experiments and changes have improved the manuscript.
Our answers are in red.
1. Page 4 line 1: I don't think people usually use the phrase "high-weight" RNA northern blots. I would suggest changing it to high molecular weight northern blots. We have changed it accordingly in the text and in the figure legend of figure 2 (P6, L8).
2. Page 8 line 38: "... the intensity the symptoms are more severe for ..." Does this mean "... the intensity of the symptoms is more severe for ..." ? Yes it was a mistake. We have changed it accordingly
Reviewer 4 Report
I have carefully read the revised manuscript and the responses given by the authors. The authors made a huge effort to answer all the gaps and they succeeded it to a satisfactory degree. Therefore, I believe that the manuscript is ready to be published.
Author Response
We are really grateful for the thorough review and all the suggestions of the reviewer. They have helped us much to improve the quality of the manuscript.